# Three-Dimensional Cephalometric Analysis of Skeletal and Dental Effects in Patients Undergoing Transpalatal Distraction

**Tomasz Żyła [1],*, Beata Kawala [2], Rafał Nowak [3] , Maciej Kawala [4] and Jowita Halupczok-Żyła [5]**

1 Private Orthodontic Practice, 55-200 Oława, Poland
2 Department of Maxillofacial Orthopaedics and Orthodontics, Wroclaw Medical University, 50-425 Wrocław, Poland; beata.kawala@umw.edu.pl
3 Department of Otolaryngology and Maxillofacial Surgery, University of Zielona Góra, 65-046 Zielona Góra, Poland; rafal.nowak@chirurgiatwarzy.pl
4 Department of Prosthetic Dentistry, Wroclaw Medical University, 50-425 Wrocław, Poland; maciej.kawala@umw.edu.pl
5 Department of Endocrinology, Diabetes and Isotope Therapy, Wroclaw Medical University, 50-367 Wrocław, Poland; jowitahz@gmail.com
* Correspondence: tomek_zyla@op.pl

**Abstract:** The present study aimed to assess dental and skeletal effects after transpalatal distraction using 3D cephalometry methodology. The study group comprised 34 patients (mean age 27.7 years) who were diagnosed with transverse skeletal maxillary deficiency of at least 7 mm. Computed tomography scans were obtained before surgical procedure (T1), after completion of expansion (T2) and at 6-month follow-up (T3). Computed tomography scans were imported into Dolphin Imaging software version 11.7 (Chatsworth, CA, USA). Three-dimensional skull models were oriented according to the Frankfurt horizontal plane, midsagittal plane (passing through the skeletal nasion) and frontal plane (passing through the right and left porion). Cephalometric landmarks related to skeletal, and dental structures were traced and linear and angular measurements were calculated. Following transpalatal distraction N-ANS and S-PNS distances increased by 1.27 mm and 0.54 mm, respectively. Skeletal expansion at the canine region (ARCR-ARCL) was 8.43 mm at T2 and 6.39 mm at T3. Expansion at the distal part of the maxilla (ARMR-ARML) was 5.95 mm at T2 and 4.81 mm after retention. The highest increase in maxillary arch width at T2 was observed at canines (8.74 mm), lower at premolars (8.33 mm) and the lowest at molars (6.76 mm). There is no anteroposterior movement of maxilla following transpalatal distraction; however, the maxilla shifts downward which is particularly marked anteriorly. Skeletal and dental expansion in the transversal plane occurs in a V-shaped manner, with more expansion at the anterior part of the maxilla.

**Keywords:** transpalatal distraction; surgically assisted rapid maxillary expansion; 3D cephalometry





## 1. Introduction

Treatment of skeletal malocclusion in individuals with completed skeletal growth represents a challenge for both the clinician and the patient, as it requires surgical release of the areas of bone resistance. Surgically assisted rapid maxillary expansion (SARME) is a widely recognized treatment modality, which employs conventional dental anchorage or, more recently, a skeletal anchorage [1]. The latter, namely: a bone-borne device (transpalatal distractor, TPD) not only eliminates side effects of using the teeth as an anchorage, but also contributes to a greater component of skeletal than dental expansion [1–6].

Maxillary expansion provokes evident skeletal and soft-tissue changes. The assessment of dental and skeletal changes involves a variety of tools, including two- and three-dimensional radiological imaging techniques [4–6]. Lateral cephalograms have inherent limitations such as image magnification, distortion, overlapping of anatomical structures and errors related to the identification of reference points. Despite these drawbacks,

2D images are still widely used in orthodontic diagnostics and monitoring of treatment progress [7]. Rapid development of computed tomography and substantial reduction in the radiation dose in modern imaging have made it possible to introduce three-dimensional craniofacial analysis [8]. Within recent years it has been proven that 3D analysis of the maxillofacial area shows a high interobserver agreement and can be a reliable and reproducible method for both clinical and research applications [9–12].

The present study aimed to assess skeletal and dental changes after transpalatal distraction, using 3D cephalometric analysis.

## 2. Material and Methods

The study was approved by the Bioethics Committee of Wroclaw Medical University and enrolled 34 patients in the study (21 women and 13 men aged 17 to 44 years), whose mean age was 27.7 years. A maxillary transverse deficiency of at least 7 mm was the main criterion for inclusion. We excluded patients with craniofacial congenital anomalies and individuals with past maxillofacial surgery. All participants signed an informed consent to participate in orthodontic-surgical treatment. The patients underwent transpalatal distraction at the Department of Maxillofacial Surgery, Wroclaw Medical University, Poland. Fixed orthodontic appliances were bonded in both jaws a few days before surgery. The maxillary archwire was sectioned between the central incisors.

### 2.1. Surgical Procedure

Under general anesthesia with nasotracheal intubation, the area of dissection was infiltrated with a local anesthetic containing a vasoconstrictor. The Le Fort I osteotomy was performed without pterygoid disjunction. The midpalatal suture was disconnected through an incision in the upper labial frenulum. The anterior, lateral and median bony supports of the maxilla were transected with a reciprocating saw and an osteotome. Mobility of the segments and the symmetry of surgical separation were tested. A transpalatal distractor (UNI-Smile, Titamed, Kontich, Belgium) with a blocking screw was placed at the level of second premolars. The blocking screw was removed after 6 days and the distractor was activated until bone resistance was felt. The patient was instructed to further activate the distractor by turning the screw twice a day (0.5 mm per day) and referred to an orthodontist for further supervision of expansion. After completion of the activation, the blocking screw was reinserted into the distractor for a consolidation period of 4 to 6 months. Subsequently, the transpalatal distractor was removed in an outpatient clinic.

### 2.2. 3D Cephalometric Analysis

Craniofacial computed tomography (CT) images were taken before (T1), immediately after TPD (T2), and 6-months post-T2 (T3) with the use of a 64-row scanner (GE Healthcare) at 0.625 mm slice thickness, with a pitch of 1.375. Tube voltage was set at 120 kV and tube current was selected individually for each patient. Craniofacial CT images in DICOM (Digital Imaging and Communications in Medicine) format were imported into the Dolphin Imaging version 11.7 software (Chatsworth, CA, USA), which allowed construction of 3D models. One investigator identified the reference points then obtained linear and angular measurements (Figures 1 and 2). To determine the intraobserver reliability, 10 randomly selected computed tomography images were analyzed for a second time (with an interval of 7 days).

### 2.3. Statistical Analysis

The statistical analysis was performed using R for Windows version 3.5. The mean, standard deviation, minimum value, maximum value, lower and upper quartiles, and median were calculated for all analyzed variables. The Shapiro–Wilk test was used to determine the normality of the data. The intraclass correlation coefficient (ICC) was used to analyze the reliability of cephalometric measurements. The data were analyzed by repeated

ANOVA or Friedman test. Post hoc comparisons were made with the Tukey method or the Nemenyi method. The level of statistical significance was set at $p < 0.05$.

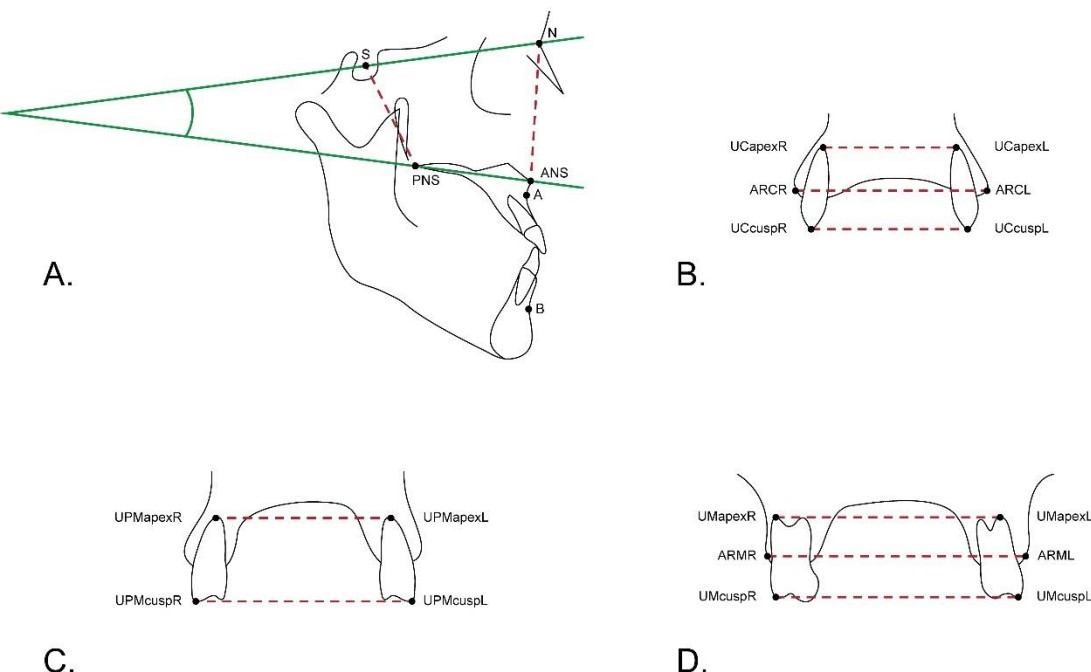

**Figure 1.** Schematic representation of the reference points and linear measurements (dashed lines) used in the study. (**A**)—sagittal landmarks and measurements, (**B–D**)—frontal landmarks and measurements. Legend. **ARM**: the most inferior and lateral point of the maxillary alveolar ridge at the first molar level; **ARC**: the most inferior and lateral point of the maxillary alveolar ridge at the canine level; **UMcusp**: the most inferior point of the mesial cusp of the first upper molar; **UPMcusp**: the most inferior point of the buccal cusp of the first upper premolar; **UCcusp**: the most inferior point of the upper canine cusp; **UMapex**: the most superior point of the mesiobuccal root of the first upper molar; **UPMapex**: the most superior point of the buccal root of the first upper premolar; **UCapex**: the most superior point of the upper canine root; **N**: the midpoint of the frontonasal suture (nasion); **S**: the center of the hypophyseal fossa (sella turcica, sella); **ANS**: the most anterior midpoint of the anterior nasal spine of the maxilla; **PNS**: the most posterior midpoint of the posterior nasal spine of the palatine bone; **A point**: the odd point located in the deepest concavity of the maxillary alveolar process, immediately beneath the anterior nasal spine; **B point**: the odd point located in the deepest concavity of the chin. R and L indicate right and left side, respectively.

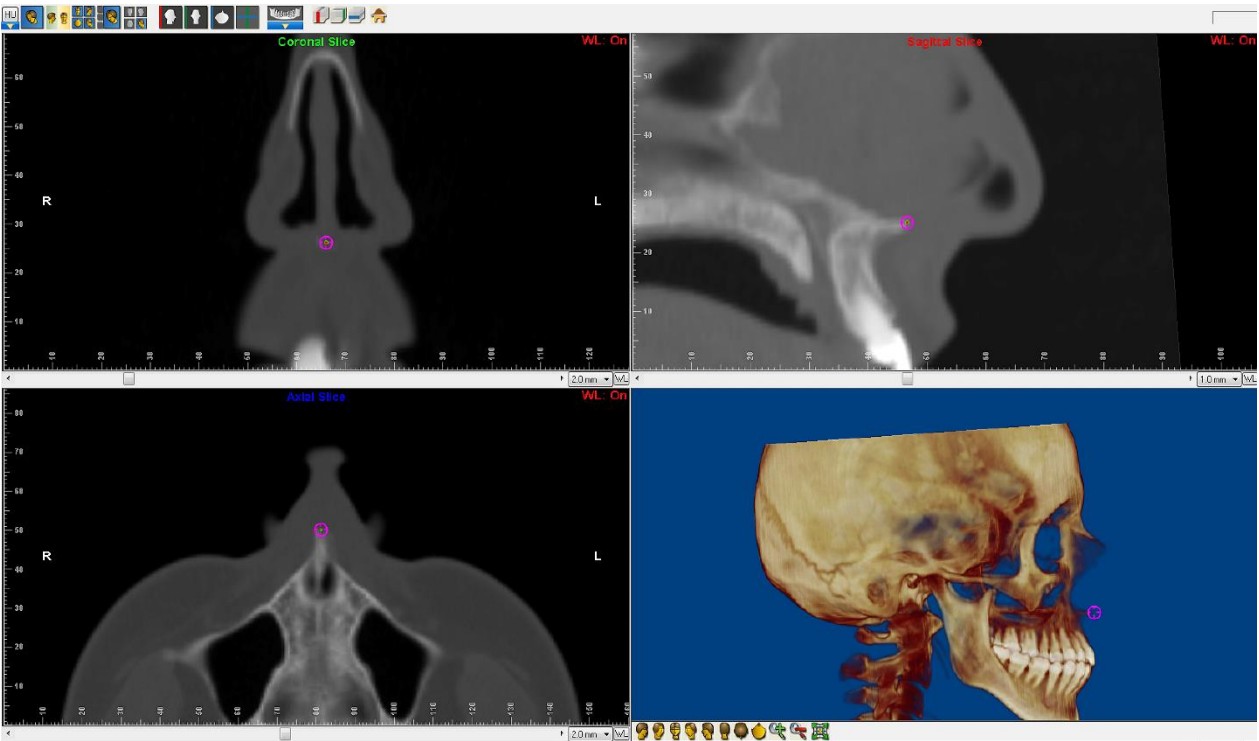

**Figure 2.** Identification of ANS reference point (anterior nasal spine) in all three planes of space.

### 3. Results

Descriptive statistics of the skeletal and dental changes within three time intervals provided as the subtraction of values T1 from T2 (2-1), T2 from T3 (3-2) and T1 from T3 (3-1) are presented in Tables 1 and 2. All linear skeletal variables underwent statistically significant changes. Vertically, the distance between the craniofacial base and the maxilla measured at its frontal (N-ANS) and posterior (S-PNS) aspects gradually increased from T1 to T3. Horizontally, the maxillary width experienced an expansion from T1 to T2: smaller at the molar (ARMR − ARML = 5.95) than at the canine level (ARCR − ARCL = 8.43), which merely slightly collapsed from T2 to T3 (Table 1). The linear changes insignificantly ($p < 0.05$) affected the angular ones.

All dental changes were statistically significant (Table 2). Immediately after TPD, the greatest width increase measured at the crown levels occurred at the canines (8.74 mm) and the smallest at the molars (6.76 mm). The most relapse was also found at the canines, less at the premolars, and the least at the molars. Except for the relapse at the canines at T3, changes of the distances between the crowns of the teeth followed the pattern of their root changes, although the extent of inter-crown expansion was greater.

**Table 1.** Descriptive statistics of the linear and angular skeletal measurements.

| Measurement | T | N | Mean T1 | Mean T2 | Mean T3 | SD T1 | SD T2 | SD T3 | Min T1 | Min T2 | Min T3 | Med T1 | Med T2 | Med T3 | Max T1 | Max T2 | Max T3 | $p^1$ | $p^2$ |
|---|---|---|---|---|---|---|---|---|---|---|---|---|---|---|---|---|---|---|---|
| N-ANS (mm) | | | 50.29 | 51.34 | 51.55 | 4.10 | 3.81 | 3.97 | 42.70 | 42.30 | 42.60 | 49.70 | 50.85 | 50.95 | 57.70 | 58.20 | 58.80 | | |
| | 2-1 | 34 | 1.05 | | | 1.28 | | | −1.20 | | | 1.35 | | | 3.10 | | | | 0.0001 |
| | 3-2 | 34 | 0.22 | | | 0.81 | | | −1.30 | | | 0.20 | | | 2.10 | | | <0.001 | 0.52 |
| | 3-1 | 34 | 1.27 | | | 1.35 | | | −1.00 | | | 1.55 | | | 4.10 | | | | 0.0001 |
| S-PNS (mm) | | | 45.14 | 45.39 | 45.68 | 4.01 | 4.15 | 4.13 | 37.10 | 37.30 | 37.80 | 46.05 | 46.60 | 46.15 | 51.70 | 52.20 | 52.60 | | |
| | 2-1 | 34 | 0.24 | | | 0.56 | | | −1.10 | | | 0.25 | | | 1.40 | | | | 0.0001 |
| | 3-2 | 34 | 0.30 | | | 0.52 | | | −0.90 | | | 0.30 | | | 1.60 | | | <0.001 | 0.01 |
| | 3-1 | 34 | 0.54 | | | 0.68 | | | −0.90 | | | 0.60 | | | 1.80 | | | | 0.045 |
| ARMR-ARML (mm) | | | 51.87 | 57.82 | 56.68 | 4.69 | 4.88 | 4.47 | 41.90 | 47.00 | 47.30 | 51.40 | 56.80 | 56.25 | 64.00 | 68.50 | 67.50 | | |
| | 2-1 | 34 | 5.95 | | | 2.15 | | | 3.10 | | | 5.30 | | | 12.20 | | | | <0.001 |
| | 3-2 | 34 | −1.14 | | | 1.11 | | | −3.60 | | | −0.95 | | | 0.30 | | | <0.001 | <0.001 |
| | 3-1 | 34 | 4.81 | | | 2.41 | | | 1.00 | | | 5.00 | | | 12.10 | | | | <0.001 |
| ARCR-ARCL (mm) | | | 35.02 | 43.45 | 41.41 | 3.58 | 3.20 | 3.92 | 25.82 | 37.60 | 34.70 | 34.15 | 43.00 | 41.80 | 42.20 | 51.30 | 50.60 | | |
| | 2-1 | 34 | 8.43 | | | 2.38 | | | 3.90 | | | 8.25 | | | 17.40 | | | | <0.001 |
| | 3-2 | 34 | −2.04 | | | 1.45 | | | −4.80 | | | −2.15 | | | 0.50 | | | <0.001 | <0.001 |
| | 3-1 | 34 | 6.39 | | | 2.85 | | | 0.20 | | | 5.55 | | | 16.70 | | | | <0.001 |
| SNA (°) | | | 82.12 | 82.06 | 82.15 | 3.19 | 3.22 | 3.20 | 77.11 | 76.92 | 77.08 | 82.10 | 82.14 | 82.16 | 90.00 | 90.30 | 90.40 | | |
| | 2-1 | 34 | 0.06 | | | 0.65 | | | −2 | | | 0.23 | | | 1.17 | | | | |
| | 3-2 | 34 | 0.09 | | | 0.29 | | | −0.3 | | | 0.01 | | | 1.01 | | | | 0.23 |
| | 3-1 | 34 | 0.15 | | | 0.50 | | | −1.12 | | | 0.12 | | | 1.06 | | | | |
| SNB (°) | | | 80.77 | 80.53 | 80.62 | 4.80 | 4.98 | 4.95 | 72.06 | 72.1 | 72.14 | 80.12 | 80.06 | 79.95 | 92.95 | 93.12 | 93.17 | | |
| | 2-1 | 34 | −0.24 | | | 0.96 | | | −4.36 | | | 0.02 | | | 0.12 | | | | |
| | 3-2 | 34 | 0.09 | | | 0.29 | | | 0.12 | | | −0.01 | | | 1.32 | | | | 0.16 |
| | 3-1 | 34 | −0.15 | | | 0.74 | | | −3.11 | | | 0.11 | | | 1.24 | | | | |

**Table 1.** *Cont.*

| Measurement | T | N | Mean | | | SD | | | Min | | | Med | | | Max | | | $p$ [1] | $p$ [2] |
|---|---|---|---|---|---|---|---|---|---|---|---|---|---|---|---|---|---|---|---|
| | | | T1 | T2 | T3 | T1 | T2 | T3 | T1 | T2 | T3 | T1 | T2 | T3 | T1 | T2 | T3 | | |
| ANB (°) | | | 1.26 | 1.53 | 1.56 | 4.07 | 4.27 | 4.31 | −11.06 | −10.95 | −11.01 | 2.06 | 2.21 | 2.14 | 8.23 | 8.17 | 8.21 | | |
| | 2-1 | 34 | 0.26 | | | 0.62 | | | −1.03 | | | 0.2 | | | 2.12 | | | | |
| | 3-2 | 34 | 0.00 | | | 0.00 | | | 0.26 | | | 0.11 | | | 0.03 | | | | 0.35 |
| | 3-1 | 34 | 0.26 | | | 0.62 | | | −1.11 | | | 0.11 | | | 2.03 | | | | |
| S-N/PNS-ANS (°) | | | 0.21 | 0.48 | 0.57 | 4.89 | 4.51 | 4.45 | −9.68 | −8.20 | −7.78 | 1.35 | 1.42 | 1.36 | 8.12 | 7.55 | 8.06 | | |
| | 2-1 | 34 | 0.27 | | | 1.04 | | | −0.80 | | | −0.05 | | | 1.35 | | | | |
| | 3-2 | 34 | 0.09 | | | 0.36 | | | −0.70 | | | 0.10 | | | 1.74 | | | | 0.27 |
| | 3-1 | 34 | 0.36 | | | 1.25 | | | −1.00 | | | 0.10 | | | 5.06 | | | | |

T—time point of measurement, N—number of patients. SD—standard deviation, Min/Max—minimum/maximum value, Med—median, $p$ [1]—ANOVA *p*-value, $p$ [2]—post-hoc *p*-value.

**Table 2.** Descriptive statistics of dental measurements.

| Measurement (mm) | T | N | Mean | | | SD | | | Min | | | Med | | | Max | | | $p$ [1] | $p$ [2] |
|---|---|---|---|---|---|---|---|---|---|---|---|---|---|---|---|---|---|---|---|
| | | | T1 | T2 | T3 | T1 | T2 | T3 | T1 | T2 | T3 | T1 | T2 | T3 | T1 | T2 | T3 | | |
| UMcuspR-UMcuspL | | | 48.09 | 54.85 | 53.19 | 5.99 | 6.16 | 5.77 | 32.40 | 40.10 | 39.40 | 48.45 | 55.00 | 53.60 | 57.30 | 69.20 | 64.80 | | |
| | T2-T1 | 34 | 6.76 | | | 3.27 | | | 2.00 | | | 6.15 | | | 16.80 | | | | **<0.001** |
| | T3-T2 | 34 | −1.66 | | | 1.41 | | | −5.20 | | | −1.25 | | | 0.50 | | | **<0.001** | **<0.001** |
| | T3-T1 | 34 | 5.10 | | | 3.14 | | | −0.50 | | | 5.40 | | | 16.90 | | | | **0.001** |
| UMapexR-UMapexL | | | 48.17 | 52.81 | 51.79 | 5.31 | 5.21 | 50.08 | 340.10 | 38.30 | 37.30 | 48.00 | 51.45 | 51.00 | 58.00 | 62.10 | 60.60 | | |
| | T2-T1 | 34 | 4.64 | | | 1.77 | | | 1.70 | | | 4.55 | | | 9.20 | | | | **<0.001** |
| | T3-T2 | 34 | −1.02 | | | 1.45 | | | −3.50 | | | −0.85 | | | 2.50 | | | **<0.001** | **<0.001** |
| | T3-T1 | 34 | 3.62 | | | 2.71 | | | −1.80 | | | 3.70 | | | 8.10 | | | | **0.004** |
| UPMcuspR-UPMcuspL | | | 38.48 | 46.81 | 44.85 | 4.46 | 4.50 | 40.47 | 300.40 | 39.90 | 37.80 | 38.20 | 45.70 | 44.55 | 51.60 | 59.50 | 58.70 | | |
| | T2-T1 | 34 | 8.33 | | | 3.09 | | | 3.00 | | | 8.30 | | | 16.60 | | | | **<0.001** |
| | T3-T2 | 34 | −1.96 | | | 1.59 | | | −6.20 | | | −1.95 | | | 0.50 | | | **<0.001** | **<0.001** |
| | T3-T1 | 34 | 6.38 | | | 2.99 | | | 2.00 | | | 6.65 | | | 14.70 | | | | **<0.001** |

**Table 2.** *Cont.*

| Measurement (mm) | T | N | Mean | | | SD | | | Min | | | Med | | | Max | | | $p^1$ | $p^2$ |
|---|---|---|---|---|---|---|---|---|---|---|---|---|---|---|---|---|---|---|---|
| | | | T1 | T2 | T3 | T1 | T2 | T3 | T1 | T2 | T3 | T1 | T2 | T3 | T1 | T2 | T3 | | |
| UPMapexR-UPMapexL | | | 34.72 | 40.77 | 39.40 | 3.99 | 4.30 | 40.52 | 270.60 | 31.70 | 31.70 | 33.65 | 41.55 | 39.30 | 40.60 | 49.40 | 47.60 | | |
| | T2-T1 | 34 | | 6.05 | | | 2.56 | | | 0.30 | | | 6.50 | | | 10.90 | | | **<0.001** |
| | T3-T2 | 34 | | −1.37 | | | 1.24 | | | −4.40 | | | −1.30 | | | 0.40 | | **<0.001** | **<0.001** |
| | T3-T1 | 34 | | 4.68 | | | 2.78 | | | −0.70 | | | 4.20 | | | 8.90 | | | **0.003** |
| UCcuspR-UCcuspL | | | 32.47 | 41.21 | 38.07 | 2.86 | 3.81 | 40.01 | 270.80 | 34.50 | 30.90 | 33.15 | 40.80 | 38.20 | 37.60 | 52.80 | 50.30 | | |
| | T2-T1 | 34 | | 8.74 | | | 2.62 | | | 4.70 | | | 8.40 | | | 17.00 | | | **<0.001** |
| | T3-T2 | 34 | | −3.14 | | | 1.69 | | | −6.10 | | | −2.85 | | | −0.40 | | **<0.001** | **<0.001** |
| | T3-T1 | 34 | | 5.60 | | | 2.76 | | | 0.20 | | | 5.95 | | | 14.50 | | | **<0.001** |
| UCapexR-UCapexL | | | 26.29 | 33.28 | 32.47 | 3.36 | 3.23 | 30.94 | 190.50 | 26.60 | 23.50 | 26.20 | 33.85 | 33.80 | 31.30 | 42.00 | 41.30 | | |
| | T2-T1 | 34 | | 6.99 | | | 1.84 | | | 3.90 | | | 7.50 | | | 10.70 | | | **<0.001** |
| | T3-T2 | 34 | | −0.81 | | | 1.64 | | | −3.90 | | | −0.85 | | | 2.60 | | **<0.001** | **<0.001** |
| | T3-T1 | 34 | | 6.18 | | | 2.71 | | | 0.70 | | | 6.00 | | | 11.30 | | | 0.130 |

T—time point of measurement, N—number of patients. SD—standard deviation, Min/Max—minimum/maximum value, Med—median, $p^1$—ANOVA $p$-value, $p^2$—post-hoc $p$-value.

## 4. Discussion

One of our study objectives was to determine the pattern of expansion and positional changes of the maxilla after TPD. We found insignificant change of the SNA, which is consistent with other studies, showing that correction of the maxillary constriction with a transpalatal distractor does not influence the anteroposterior position of the A point [1,13]. On the other hand, Parhiz et al. reported a post-TPD increase of the SNA angle by $1.6° \pm 2.57°$ ($p = 0.0001$); however, such standard deviation value proves the high variability of the outcomes. Moreover, 20-month follow-up and an orthodontic closure of the diastema may also affect the TPD results, because remodeling of the alveolar bone between the maxillary central incisors is likely to influence position of the A point, and thus the SNA angle [14].

Neither position of the mandible (SNB angle) nor inclination of the maxilla to the cranial base (SN/PNS-ANS angle) changed after our surgical protocol. Nonetheless, although this is in accordance with recently published results [13,14], it must be emphasized that studies on TPD effects give contradictory results. Hansen et al. described counter clockwise mandibular rotation, while other authors observed the opposite [14–16]. Parhiz et al. suggest that an increase in the inclination of the maxilla to the cranial base induced by TPD can be only expected when the S-N/PNS-ANS angle is pre-operatively low [14]. Our study seems to support this hypothesis, as the discussed angle was low at T1, and changed insignificantly at T2 and T3. Except for increasing the S-N/PNS-ANS angle, the whole maxilla moved downward: the N-ANS and the S-PNS measured at T3 significantly increased by 1.27 mm and 0.54 mm, respectively, compared with T1. This is in accordance with the majority of researchers' observations [1,13,14]. To us, two factors seem to be of key importance for vertical control: 1. position of the distractor in relation to the center of resistance of the mobilized maxilla and 2. whether the pterygomaxillary disjunction is performed or not. The TPD (or the bone-borne SARME) generates a force vector located close to the center of resistance of the maxilla, which ensures more parallel movement of the separated segments. During expansion, one observes buccal movement of the maxillary halves, which results in downward rotation of the structures located close to the cranial midline, including A, ANS, and PNS points. This rotation increases together with elongation of the force moment arm, which is supported by the study of Koudstaal et al., who—comparing results of the tooth-borne Hyrax and TPD application—observed a greater downward movement of the maxilla resulting from the previous appliance [1].

Greater mobility of the separated segments at the anterior part of the maxilla and—in turn—more pronounced downward movement in the canine region may also be explained by the lack of pterygomaxillary disjunction in our study. This way our TPD protocol results resemble the Hyrax screw-induced pattern. This is in accordance with the report of Lee et al., who—using finite element analysis—showed that pterygoid disjunction provokes an even downward movement of both the front and the lateral aspects of maxilla, measured at the level of the midpalatal suture, whilst alteration the surgical protocol by omitting the disjunction and applying merely the Hyrax appliance produces downward displacement that is greater anteriorly. These results seem to confirm the fact that the extent of the surgical osteotomy affects the pattern of maxillary displacement [17].

We achieved transverse skeletal changes that were greater at the anterior part of the maxilla: the ratio of anterior to posterior skeletal expansion measured at the level of the canine (ARC) and the first molar roots (ARM) was 1.33 at T3. The pattern of skeletal expansion was V-shaped, with the base at the area of the anterior nasal spine, and the apex at the distal part of the palatal suture. Similar effects are observed for the RME with Hyrax screw, and with both bone-borne and tooth-borne SARME, due to their comparable anteroposterior location between the second premolars and the first molars [3,6,15,18–21]. In patients with posterior constriction of the maxilla, a parallel rather than V-shaped expansion would be of greater benefit. This may be achieved either by shifting the force vector more distally, e.g., placing the TPD at the level of the molars or by shifting the center of resistance of the maxilla anteriorly, namely: by performing pterygoid disjunction [3,4,22].

As for the dental changes, they corresponded to the skeletal ones with the greatest increase in the canine area. The changes of the inter-root distances were smaller than those measured at the crown levels. This is a confirmation that the maxillary segments tilt buccally as a result of TPD. In addition, since the dental/skeletal expansion ratio was close to 1.0 at the canine level, this suggests that the expansion mainly results from translation of the maxillary segments. In the molar region, the discussed ratio reached 1.14 indicating that the increase of the intermolar width was the result of a combination of bodily and tipping movements of the maxilla. In our study, immediately after transpalatal distraction (T2) the ratio of canine to molar dental expansion was 1.29, which is close to the finding (1.5) reported by Pinto et al. and Ramieri et al. using similar methodology [4,5].

As for dental relapse, we found significant changes in all studied dental measurements. The greatest decrease in the dental width at T3 was observed between the canines (3.14 mm), and smaller between the premolars (1.96 mm), and the molars (1.66 mm). The decrease in the premolar inter-root width was 1.37 mm ($p = 0.003$), and the molar inter-root was 1.02 mm ($p = 0.004$). However, the canine inter-root distance at T3 increased by 0.81 mm, although this change was not statistically significant ($p = 0.13$). This suggests that the inter-canine width reduction resulted from palatal tipping of the crowns with simultaneous buccal root torque. To us, several factors may account for these changes. Due to the interdigitation of teeth in the distal part of the dental arch, the maxillary premolars and molars are stabilized in occlusive contact by the lingual cusps of the mandibular teeth. The canines can move more freely into the palatal direction than the lateral teeth. Furthermore, during consolidation period followed by the treatment with fixed appliances, the transpalatal distractor acts as a stabilizing appliance providing both: increased retention in the distal part of the maxilla (where the TPD is located) and undisturbed, spontaneous migration of the anterior teeth towards the midline.

*Limitations*

Contemporary protocol of the maxillary expansion with TPD requires mounting the fixed appliance prior to the surgical intervention. Therefore, dental changes following TPD application, especially those occurring at T3, display the net movement resulting from the maxillary halves displacement during expansion and from an orthodontic treatment. Additionally, even though this may have biased our results in terms of solely TPD-induced dental changes, it is out of the question that our study provides valuable clinical data to aid an interdisciplinary orthodontic-surgical treatment.

## 5. Conclusions

1. The clear advantage of 3D analysis over 2D analysis is that the former allows the determination of changes in all planes of space using only one radiological exposure.
2. TPD does not produce a significant anteroposterior displacement of the maxilla. However, factors such as surgical technique, distraction protocol, as well as individual anatomical differences may have an impact on the outcome.
3. TPD causes transverse and vertical changes useful in the treatment of an open bite caused by maxillary constriction and anterior rotation of the palatal plane.
4. If the SARME protocol does not involve pterygoid disjunction and the TPD is placed at the level of second premolars, more skeletal and dental expansion occurs at the anterior than at the posterior part of the maxilla, which may be beneficial in the treatment of anterior transverse maxillary deficiency.

**Author Contributions:** Conceptualization, T.Ż., B.K. and R.N.; Methodology, T.Ż., B.K., R.N. and M.K.; Validation, B.K., M.K. and J.H.-Ż.; Formal Analysis, T.Ż.; Investigation, T.Ż.; Resources, T.Ż. and R.N.; Writing—Original Draft Preparation, T.Ż.; Writing—Review and Editing, B.K. and J.H.-Ż.; Visualization, T.Ż. and J.H.-Ż.; Supervision, B.K., R.N. and M.K. All authors have read and agreed to the published version of the manuscript.

**Funding:** This research received no external funding.

**Institutional Review Board Statement:** The study was conducted in accordance with the Declaration of Helsinki, and approved by the Ethics Committee of Wroclaw Medical University (KB—261/2015, 15 May 2015).

**Informed Consent Statement:** Patient consent was waived due to the retrospective nature of the study.

**Data Availability Statement:** Not applicable.

**Conflicts of Interest:** The authors declare no conflict of interest.

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
