# Peer review of "Three-Dimensional Cephalometric Analysis of Skeletal and Dental Effects in Patients Undergoing Transpalatal Distraction"

_applsci, doi:10.3390/app12094273_

Round 1

Reviewer 1 Report

Thank you for the opportunity to review this paper. I enjoyed reading. In general, it is well written and the results presented well. The data seems to support what it currently in the literature.

My only clarification/concern is - were the patients exposed 3 times to the CT scan? Was there any other information gathered and what might the collective radiation dose be?

Author Response

1) My only clarification/concern is - were the patients exposed 3 times to the CT scan? Was there any other information gathered and what might the collective radiation dose be?

We are grateful for the Reviewer’s positive response to the paper.

Each patient included in the study had 3 CT scans: T1 - before TPD, T2 - immediately after TPD, and T3 – at a 6-month follow-up. We did not gather any additional data. We only used information extracted from the CT scans. The collective radiation dose was approximately 6.5 mSv (DLP = 2100 mGy cm).

Reviewer 2 Report

The article is well prepared and results are interesting. However I have couple of questions:

How were performed Le Fort I osteotomie and midpalatal dissection? By saw, piezo surgery device, burr?

Does some measurements interference with metal artefact occur? If any, how were there adressed?

Do the autors anticipate some future maxilla dimesion changes in youger patients related to bone growth?

Author Response

We would like to thank Reviewer 1 for the thorough reading of this manuscript and valuable comments, which helped us to improve the quality of the paper.

  • How were performed Le Fort I osteotomie and midpalatal dissection? By saw, piezo surgery device, burr?

The LeFort I osteotomy and midpalatal dissection were performed using a reciprocating saw and an osteotome. We have added that information to the Materials and Methods section (lines 59-60).

  • Does some measurements interference with metal artefact occur? If any, how were there adressed?

We did not experience any difficulties in locating the reference points. The presence of fixed appliances and transpalatal distractors did not produce artifacts that interfered with landmark identification.

  • Do the autors anticipate some future maxilla dimesion changes in youger patients related to bone growth?

We have found reports of TPD (or SARME) being used in children at the age of 14 years. In that case, it is safe to presume that some amount of growth can occur after the completion of orthodontic and surgical treatment. The youngest patient in our study group was 17 years old. The mean age was 27.7 years old. We do not think that the maxillary dimensions will significantly change in our patients.